# Improved gRNA secondary structures allow editing of target sites resistant to CRISPR-Cas9 cleavage

Stephan Riesenberg [1✉], Nelly Helmbrecht [1], Philipp Kanis [1], Tomislav Maricic [1] & Svante Pääbo [1,2]

The first step in CRISPR-Cas9-mediated genome editing is the cleavage of target DNA sequences that are complementary to so-called spacer sequences in CRISPR guide RNAs (gRNAs). However, some DNA sequences are refractory to CRISPR-Cas9 cleavage, which is at least in part due to gRNA misfolding. To overcome this problem, we have engineered gRNAs with highly stable hairpins in their constant parts and further enhanced their stability by chemical modifications. The 'Genome-editing Optimized Locked Design' (GOLD)-gRNA increases genome editing efficiency up to around 1000-fold (from 0.08 to 80.5%) with a mean increase across different other targets of 7.4-fold. We anticipate that this improved gRNA will allow efficient editing regardless of spacer sequence composition and will be especially useful if a desired genomic site is difficult to edit.

[1] Max Planck Institute for Evolutionary Anthropology, Leipzig, Germany. [2] Okinawa Institute of Science and Technology, Onna-son, Japan.
✉email: stephan_riesenberg@eva.mpg.de

The CRISPR-Cas9 system is an invaluable tool for genome modification. It uses the enzyme Cas9 that introduces a DNA double-strand break (DSB) in a target DNA sequence that is complementary to a 20-nt 'spacer sequence' in a guide RNA (gRNA) that is non-covalently bound to the enzyme. A further requirement is that a 'protospacer adjacent motif' (PAM) is present adjacent to the genomic target sequence. The gRNA can be provided as a duplex of spacer containing CRISPR RNA (crRNA) and a partially complementary trans-activating crRNA (tracrRNA). Alternatively, a single gRNA (sgRNA) can be used where crRNA

and tracrRNA are linked by an artificial loop[1] (Fig. 1a). Each gRNA thus consists of a target-specific spacer sequence and a constant part comprised of distinct conserved motifs including the nexus, and two hairpins[2]. Cellular repair of targeted DNA DSBs by error-prone end-joining pathways often results in frameshift mutations that can be used to knockout genes of interest while homology-directed repair (HDR) allows precise introduction of substitutions present in DNA molecules introduced into the cells.

Chemical modifications are often introduced in gRNAs to improve their stability against degradation by cellular nucleases.

**Fig. 1 Genome editing efficiencies of gRNAs with an engineered tracrRNA backbone. a** gRNA (hybridized crRNA:tracrRNA) from *S. pyogenes* for a canonically folded gRNA; and **b** a predicted non-canonical folded gRNA (*SSH2*). The spacer sequence, as well as the structural motifs nexus, 1st hairpin, and 2nd hairpin are shown. **c** Locked tracrRNA (t-lock) gRNA design carrying an elongated 1st hairpin with a superstable loop that should serve as a gRNA folding nucleation site and prevent misfolding. **d** Genome editing efficiencies for t0 (normal), t-lock (red), and commercial t0 (IDT) (black frame). t0 (IDT) has the same nucleotide sequence as t0, but carries additional proprietary chemical modifications. Independent biological replicates are depicted by black dots (n = 3, except *PIGZ* t0 n = 2) and the error bars show the SEM. The panels show the efficiencies for 10 different gRNAs with different predicted non-canonical spacer interactions to the nexus (cyan), 1st hairpin (pink), 2nd hairpin (blue), and/or the spacer sequence itself (orange). The predicted crRNA:tracrRNA structure from the RNA structure web server[23] is shown below the spacer sequence. The folding predictions with the locked tracrRNA are identical except for an elongated 1st hairpin and the formation of a 1st hairpin in the case of *OSBP2* and *C3*. **e** Box plots of relative genome editing efficiencies for gRNAs from d (n = 10) with respect to the corresponding t0 tracrRNA set to 100%. Boxes extend from the 25th to 75th percentile and show the median as a line. Whiskers extend from the minimum to the maximum values. Chemically synthesized gRNAs (hybridized crRNA:tracrRNA) were lipofected into Cas9 expressing 409-B2 iCRISPR human induced pluripotent stem cells (hiPSCs). Source data are provided as a Source data file.

These include 5′- and 3′-end protection using phosphorothioate bonds and internal 2′-OMe residues[3,4]. Genome editing efficiency can be further increased by so-called 'non-homologous oligonucleotide enhancement' (NOE) when an external non-homologous DNA is provided which may divert the cells towards error-prone instead of error-free repair pathways[5]. This can also be achieved by using DNA as electroporation enhancer or a DNA donor as a template for HDR.

Another approach to improve genome editing efficiency is to extend the length of complementary sequences where crRNA and tracrRNA hybridize with each other, which possibly increases the binding of Cas9 to the sgRNA[6]. If sgRNAs are expressed in vivo from a U6 promoter, a TTTT motif in the stem of the sgRNA acts as transcription termination signal. This can be overcome by swapping opposing bases of one T-A base-pair in the motif. These improvements can be combined into what is called 'hybridization extended A-T inversion' (HEAT) sgRNA. In addition, gRNA algorithms have been developed to design gRNAs in silico but their performance is far from satisfying[7]. However, in spite of these improvements, genome editing efficiencies vary strongly from target to target. In fact, some targets are intractable to genome editing even when chemically stabilized gRNAs, HEAT sgRNAs, and NOE are used.

It is likely that gRNA misfolding may reduce editing efficiency either directly or through competition for binding to the Cas9 enzyme between misfolded gRNAs and active gRNAs[8]. In agreement with this, a machine-learning model trained on activity data from over 50,000 sgRNAs revealed that self-folding free energy strongly influences cleavage efficiency and impacts the model output[9]. Further, a PAM-proximal GCC-motif has been described to block editing[10]. Attempts to break unwanted non-canonical interaction with a base substitution in the sgRNA backbone can be successful[8] but this approach has not been generalized due to the inaccuracies of RNA-folding prediction[11] and the need to design and validate novel gRNA backbone base substitutions for each new gRNA target.

Here, we present an approach that overcomes low or absent Cas9 cleavage efficiencies by the introduction of a highly stable hairpin and chemical modifications and show that this 'Genome editing Optimized Locked Design' (GOLD)—gRNA allows robust genome editing regardless of spacer sequence composition.

## Results

**Modified gRNA backbone designs**. We hypothesized that the introduction of a highly stable hairpin into the gRNA would provide a nucleation site for RNA folding and thus prevent misfolding of gRNAs (Fig. 1a, b) regardless of the spacers sequence. Consequently, we designed a tracrRNA in which the first hairpin 3′ of the nexus is elongated with a RNA hairpin with a melting temperature of 71 °C[12], which we name 'locked' hairpin tracrRNA (Fig. 1c).

We used a human induced pluripotent stem cell (hiPSC) line (409-B2) carrying a doxycycline-inducible Cas9 (iCRISPR-Cas9) gene[13] to test the efficiency of normal tracrRNA (t0), the locked tracrRNA (t-lock), and the normal tracrRNA with proprietary chemical modifications (t0-IDT) from a commercial provider in combination with ten different crRNAs, most of which have predicted strong non-canonical interactions to different regions in the gRNA (Fig. 1d). The Cas9 expressing cells were lipofected with the chemically synthesized gRNAs for 24 h. Five days later, DNA was extracted, target sequences were amplified by PCR, and the amplification product was sequenced to determine genome editing efficiencies. The locked tracrRNA increased editing efficiencies relative to the normal tracrRNA for eight of ten targets irrespective of the position of the predicted non-canonical

interaction (Fig. 1d). On average, the locked tracrRNA increased the cleavage efficiency in the targets to 169% (range 75–262%) (Fig. 1e). By comparison, the tracR with proprietary chemical modifications increased editing efficiency to an average of 131% (range 80–169%) relative to the normal tracrRNA.

To further investigate the efficacy of locked hairpins in different gRNA delivery systems and formats, we used synthesized double-stranded DNA molecules carrying T7 or U6 promoters followed by sequences coding for sgRNAs with HEAT modifications carrying locked hairpins at different positions (Supplementary Fig. 1a). Chemically synthesized crRNA:tracrRNA duplex gRNAs, sgRNAs transcribed in vitro from the T7 promoters, or DNA templates with U6 promoters that allow in vivo eukaryotic transcription of sgRNAs were electroporated into Cas9 expressing cells. The hairpin lock in the 1st hairpin increased relative cleavage efficiencies for chemically synthesized crRNA:tracrRNA duplex gRNA, in vitro transcribed (IVT) sgRNA, as well as in vivo transcribed sgRNA from linear double-stranded DNA or plasmids (Supplementary Fig. 1b). The addition of a hairpin lock in the sgRNA fusion loop did not change cleavage efficiency. Further addition of a locked hairpin at both the 3′ and 5′ almost abrogated cleavage, while addition at the 3′ end decreased the efficiency of the IVT sgRNA. We also tested other locked designs that carry hairpin loop sequences described to form super-stable, non-canonical stabilizing interactions[14] (UUCG), (CUUG), or (GCAA) (Supplementary Fig. 2a) and find that they all increased genome editing efficiency (Supplementary Fig. 2b).

In a different approach, we attempted to engineer a tracrRNA that would break unwanted non-canonical interactions that may occur when using an unmodified tracrRNA sequence. We speculated that increasing the sequence difference to the unmodified tracrRNA sequence could generate an alternative tracrRNA, which could be used for spacers that form non-canonical interactions with the unmodified tracrRNA. To this end, we designed five different tracrRNAs where we swapped opposing bases in the nexus, 1st hairpin, and 2nd hairpin at position that does not interact with the Cas9 protein. The cleavage efficiencies of the corresponding crRNA:tracrRNA gRNAs were then tested in Cas9 expressing cells as described above (Supplementary Fig. 3a, b). Nexus modification strongly reduced genome editing efficiency for all targets, while modifications in the hairpins often reduced and only rarely increased genome editing efficiency (Supplementary Fig. 3c).

**Improved chemical modifications**. To further improve editing efficiency we explored chemical modifications of gRNAs using phosphorothioate bonds for the terminal nucleotides and internal 2′OMe modifications.

The crystal structure of Cas9 with bound gRNA and target DNA (NDB: 5F9R)[15] shows that 2′OH groups of nucleotides in the nexus form polar contacts within the nexus. We hypothesized that this could stabilize the active state of the Cas9 ribonucleoprotein (Fig. 2a, b) and consequently, chemical modification (Fig. 2c) of these positions might be detrimental. To evaluate if the nexus is unsuitable for chemical modifications, we tested a medium efficiency crRNA duplexed with a tracrRNA carrying phosphorothioate end-protection at both ends (t0*), phosphorothioate end protection as well as 2′OMe (t-2′OMe*) adapted from Yin et al.[4] in which we did not modify a nexus-proximal uracil that interacts with Arg75 and Tyr72 of the Cas9 bridge helix, which has been described to modulate DNA cleavage[16], and the latter but without 2′OMe modifications in the nexus loop (t-2′OMe-2.0*) (Fig. 2d). The absence of the 2′OMe modifications of the nexus loop increases absolute genome editing efficiency from

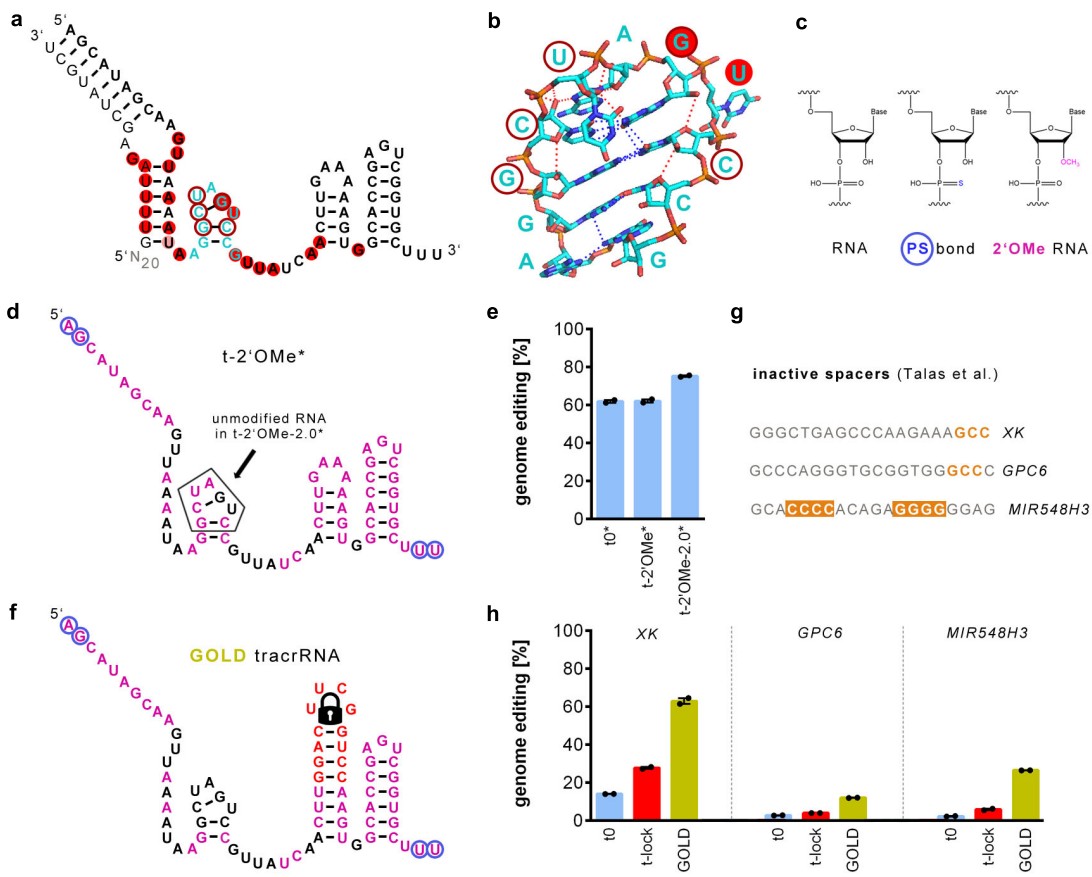

**Fig. 2 Optimization of chemical modifications of the gRNA. a** Nucleotides of the gRNA (hybridized crRNA:tracrRNA) of which the 2'OH interacts with Cas9 or with residues of the nexus (cyan) are underlaid in red or circled in dark red, respectively. The residue underlaid in rose interacts with both Arg75 and Tyr52 of the bridge helix that modulates DNA cleavage. **b** Close-up of the crystal structure of the nexus and interacting residues adapted from NDB: 5F9R. Polar interactions are shown as dotted lines and those of 2'OH residues are shown as dotted red lines. **c** Single RNA nucleotide structure and chemical modifications. **d** Chemical modifications of the tracrRNA with phosphorothioate bonds and 2'OMe modifications (t-2'OMe*). The nexus loop of t-2'OMe-2.0* only contains unmodified RNA nucleotides (colored according to c). t0* only contains phosphorothioate bonds. **e** Genome editing efficiency of different chemically modified tracrRNAs hybridized with a medium cleavage crRNA electroporated into Cas9 expressing 409-B2 iCRISPR (hiPSCs). **f** Structure and chemical modifications of the genome editing optimized locked design (GOLD) tracrRNA. **g** Spacer sequences of previously described inactive gRNAs. Putative cleavage-inhibiting motifs are highlighted orange. **h** Genome editing efficiency of normal tracrRNA (blue), t-lock tracrRNA (red), and GOLD tracrRNA (gold) hybridized with crRNAs from **g** after electroporation into Cas9 expressing 409-B2 iCRISPR hiPSCs. Independent biological replicates (n = 2) are depicted by black dots and the error bars show the SEM. Source data are provided as a Source Data file.

62 to 75%. Compared to only phosphorothioate end-protection no increase was achieved when the nexus carried 2'OMe modifications (Fig. 2e). Thus, chemical modification of a gRNA by phosphorothioate end-protection and internal 2'OMe modifications that do not include the nexus loop increases genome editing efficiency.

We combined the unmodified locked hairpin with the optimized chemical modifications of the gRNA into a 'Genome editing Optimized Locked Design' (GOLD)—tracrRNA (Fig. 2f), and tested this optimized tracrRNA in combinations with three crRNAs containing spacers that were unable to cleave their targets in human cells as well as in *E.coli* cells[17]. Two of these spacers contain a PAM-proximal GCC motif, which has been described to abrogate cleavage[10] (Fig. 2g). The GOLD-tracrRNA increased editing efficiencies for all three spacers 6.8-fold (range 4.3–11.5-fold) relative to the tracrRNA without any improvements and 3.2-fold (range 2.3–4.5-fold) relative to the locked tracrRNA without chemical modifications (Fig. 2h). In absolute terms, mean genome editing efficiency increased from 6 to 34% of chromosomes relative to the standard tracrRNA. We also tested the different tracrRNA designs with four other targets using Cas9

ribonucleoprotein (RNP) electroporation in HEK293, K562, Jurkat, Caco-2, SNL 76/7 (mouse), or CHO-K1 (chinese hamster) cells and achieved comparable increases in efficiency (Supplementary Fig. 4). In the most extreme case, in CHO-K1 cells, the editing efficiency increased 1006-fold from 0.08 to 80.5%.

Next, we investigated whether more efficient editing efficiency at the intended (on-) target by the GOLD-tracrRNA might result in increased editing of unintended targets that have sequence similarity to the on-target site. We chose six gRNA spacers that differ in their predicted tendency to result in off-target editing based on the cutting frequency determination (CFD) score[18,19] which ranges from 0 (unlikely off-target) to 1 (very likely off-target). The off-target scores of our gRNAs range from 0.2 to 1 (Supplementary Fig. 5a). After Cas9 RNP editing in 409-B2 hiPSCs with cRNA hybridized to either the unmodified tracrRNA or the GOLD-tracrRNA, we sequenced the on-target site as well as the top two predicted off-target sites for each of the six gRNAs. The GOLD-tracrRNA increased on-target editing efficiency for the six spacers relative to the unmodified tracrRNAs, but also increased the efficiency of editing of the two off-targets site with the highest CFD scores (0.91 and 1) from around 0.5 to 9%

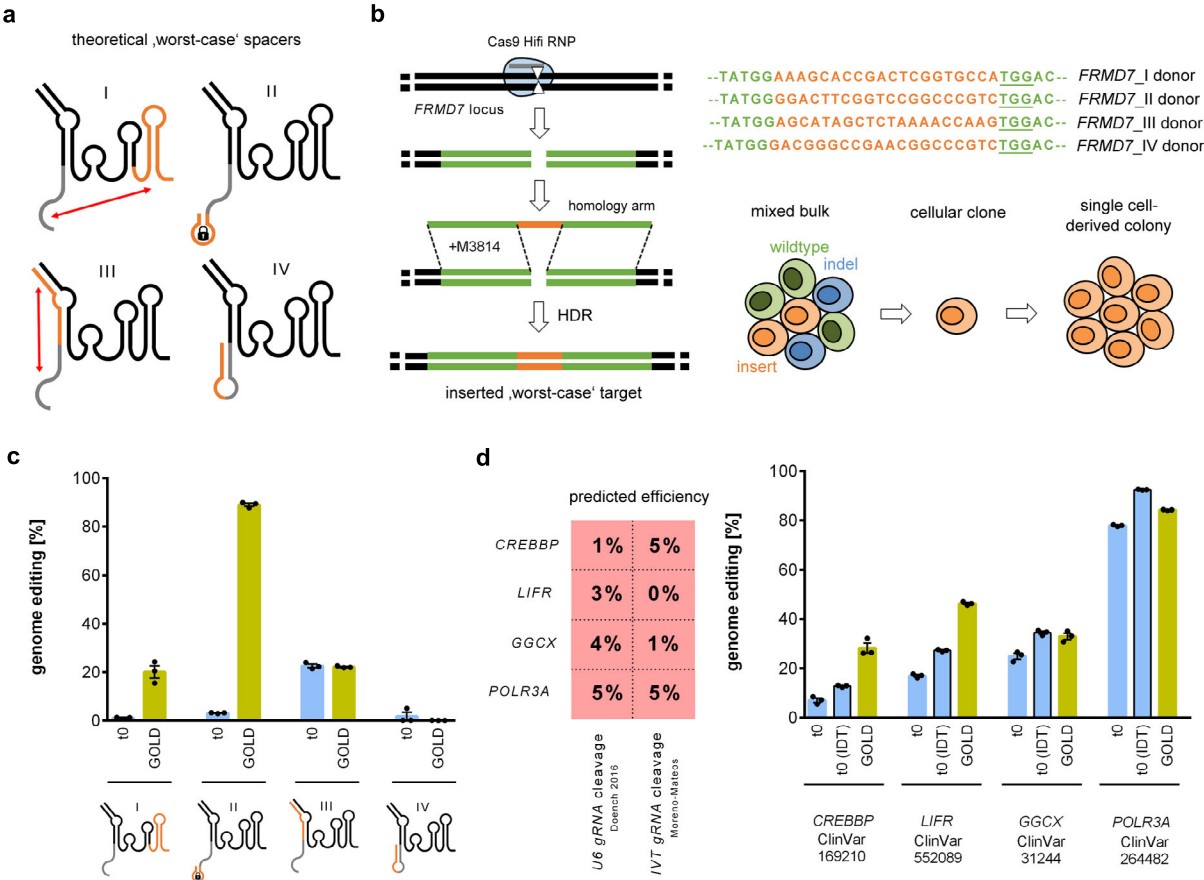

**Fig. 3 Genome editing efficiencies of predicted low performance gRNAs. a** Sketches of theoretical 'worst-case' spacers where the spacer is perfectly complementary to the 3′ end of the tracrRNA (roman numeral I), forms the locked hairpin with itself (roman numeral II), is perfectly complementary to the 3′ end of the crRNA (roman numeral III), or forms a perfect hairpin along the whole spacer sequence (roman numeral IV). The complementary region is colored orange and expected interactions are presented as hairpins or indicated by red arrows. **b** Strategy to introduce the theoretical 'worst-case' spacer target sequences from c into 409-B2 iCRISPR hiPSCs. Cleavage of the FRMD7 locus by Cas9 HiFi RNP is followed by subsequent HDR using a ssODN donor sequence carrying the intended target site next to a TGG PAM. M3814 is added to block NHEJ and increase HDR efficiency. Four different edits are done for target insertions I–IV followed by generation of single cell-derived cellular clones. **c** Genome editing efficiency of theoretical 'worst-case' gRNAs from a and t0 or GOLD tracrRNA. gRNAs were electroporated into Cas9 expressing 409-B2 iCRISPR hiPSCs that carry the corresponding inserted 'worst-case' target in the FRMD7 locus. **d** Genome editing efficiency of gRNAs adjacent to a disease-relevant ClinVar site and predicted to be self-complementary and to have very low cleavage efficiency (Doench 2016 and Moreno-Mateos percentile score both ≤5). Compared are t0 (normal), t0 (IDT) with proprietary chemical modifications, and GOLD tracrRNA. Cas9 RNPs were electroporated into 409-B2 hiPSCs. Independent biological replicates (n = 3) are depicted by black dots and the error bars show the SEM. Source data are provided as a Source Data file.

(Supplementary Fig. 5b). Another off-target with a CFD score of 0.72 increased from 0.15 to 0.26%. The other nine off-targets sites with CFD scores ranging from 0.2 to 0.78 were not edited more efficiently by the GOLD-tracrRNA. Importantly, when we edited the same targets using a high fidelity version of Cas9 with the R691A mutation (HiFi Cas9)[20] the GOLD-tracrRNA increased on-target editing for all six spacers, while off-target editing at all 12 off-target sites was unchanged with respect to editing with the unmodified tracrRNA (Supplementary Fig. 5c).

**Predicted bad loci and worst-case targets**. To test the robustness of the GOLD-tracrRNA, we chose four 'worst-case' DNA targets, for which the corresponding gRNA spacer: (I) is perfectly complementary to the 3′-end of the tracrRNA; (II) forms a locked hairpin[12] with itself and carries an inhibitory PAM-proximal GCC motif[10]; (III) is perfectly complementary to the 3′-end of the crRNA; or (IV) forms a perfect hairpin along the whole spacer sequence (Fig. 3a). We introduced each of these targets into the

FRMD7 gene of Cas9 expressing cells by genome editing using the relevant DNA donors for HDR followed by clonal isolation of cells carrying the target (Fig. 3b). We then electroporated corresponding gRNAs (hybridized crRNA:tracrRNA) into the respective cell lines. The GOLD-tracrRNA changed the editing efficiencies from 1.2% with the standard tracrRNA to 20.1% for target I, from 3.2 to 89% for II, from 22.6 to 22.2% for III, and from 1.9 to 0.2% for target IV (Fig. 3c).

We also tested the efficiency of the GOLD-tracrRNA relative to the commercially available chemically modified tracrRNA (t0-IDT) by targeted cleavage adjacent to disease-associated positions from a database for human variations and phenotypes (ClinVar)[21] associated with Rubinstein-Taybi syndrome (CREBBP), Stueve-Wiedemann syndrome (LIFR), pseudoxanthoma elasticum-like disorder (GGCX), and Wiedemann-Rautenstrauch syndrome (POLR3A). The gRNAs for these targets are predicted to have low cleavage efficiency (≤5% percentile for both the Doench et al. 2016[18] and the Moreno-Mateos et al. prediction scores[22]) and to be self-complementary by an RNA-

folding algorithm[23]. After Cas9 ribonucleoprotein (RNP) electroporation and sequencing of the targets, the editing efficiency with the normal tracrRNA is below 25% for three gRNAs and can thus be considered inefficient, while it is 78% for the fourth target, illustrating the inaccuracy of the current prediction algorithms (Fig. 3d). t0-IDT, as well as GOLD-tracrRNA, increased the editing efficiency for the four targets relative to the normal tracrRNA 1.5-fold and 2.3-fold, respectively. The GOLD-tracrRNA performed approximately two-fold better than the t0-IDT for the two least efficient gRNAs and has a comparable or only slightly lower efficiency for the other two gRNAs.

**In vitro binding and cleavage**. To investigate the extent to which the GOLD-tracrRNA design affects DNA target binding we used the four 'worst-case' DNA targets as well as the genomic *FRMD7* target, for which we achieve efficient editing with the unmodified tracrRNA. We used synthetic 90 bp-DNA molecules carrying the target sequences with adjacent genomic sequence and performed electrophoretic mobility shift assays in agarose by incubating target DNA together with RNPs consisting of catalytically dead (d)Cas9 and respective gRNAs for 16 h (Supplementary Fig. 6a, b). The GOLD-tracrRNA increases the amount of target DNA bound by dCas9 at least two-fold relative to the unmodified tracrRNA for the genomic *FRMD7* target and 'worst-case' targets I, II, and IV (Supplementary Fig. 6d).

To investigate the extent of target site cleavage we used the same synthetic targets as above which upon cleavage will yield two cleavage products of 60 and 30 bp. We separated and visualized the products in agarose gels (Supplementary Fig. 6a, c) after incubation with Proteinase K to eliminate band shifting by bound Cas9 and RNase A to remove bands arising from partially double-stranded gRNA. In agreement with previous observations[2], in vitro cleavage efficiencies differ from efficiencies achieved in cells for some gRNAs. However, the GOLD-tracrRNA changed the in vitro cleavage efficiency from 50.3% with the standard tracrRNA to 70% for *FRMD7*, from 1.6 to 3.1% for I, from 56.3 to 65.6% for II, and from 2.8 to 2.2% for target IV (Supplementary Fig. 6e).

## Discussion

Unintended secondary structures in gRNAs contribute to low genome editing efficiencies. For example, hairpin structures in the spacer portion of the gRNA or interactions between the spacer portion and the backbone of the gRNA will render gRNAs inactive[8]. Such unwanted interactions can be broken by nucleotide substitutions in the interacting sequences[8]. However, this requires different gRNA backbone substitutions for every spacer sequence of interest. Here we provide a generalizable solution to unintended intramolecular gRNA interactions by elongating the 1st hairpin of the tracrRNA with the C(UUCG)G loop motif which is extremely stable[12,24]. The 1st hairpin is suitable for this modification because it tolerates elongation[25] and can even be eliminated with only little loss of gRNA activity[2]. The modified, 'superstable' or 'locked' 1st hairpin provides an efficient nucleation site for RNA folding and therefore prevents misfolding of the tracrRNA.

TracrRNAs carrying locked hairpins increase genome editing efficiencies for eight of ten targets tested irrespective of the position of the predicted non-canonical interaction (Fig. 1). They do so irrespective of if the gRNAs are chemically synthesized, transcribed in vitro or in vivo, irrespective of if the oligonucleotides are introduced into cells by lipofection or electroporation (Fig. 1 and Supplementary Fig. 1), and irrespective of cell type or species (Supplementary Fig. 4). Different RNA loops known to be highly stable can all be used to increase genome editing efficiency

when added to the 1st hairpin (Supplementary Fig. 2), indicating that it is the stability of the secondary structure formation per se that increases editing efficiency.

We combined the improved tracrRNA design with the introduction of terminal phosphorothioate and internal 2′OMe modifications[3,4] into the 'GOLD-tracrRNA'. This design allowed several difficult-to-edit targets to be successfully edited (Fig. 2). The only case in which the GOLD-tracrRNA did not improve editing efficiency was when the spacer sequence was fully self-complementary. Fortunately, this is extremely rare. Nevertheless, even such a target might be possible to edit if a mismatch to the target sequence that prevents self-folding (but still allow some cleavage) is introduced in the spacer.

When introducing chemical modifications, we considered the fact that the nexus part of the tracrRNA is highly conserved among bacteria[2] and that 2′OH groups of nucleotides in the nexus exhibit direct polar contacts within the nexus (Fig. 2b). We, therefore, avoided the introduction of 2′OMe modifications in the nexus. Indeed, for the target tested, editing efficiency was higher when the nexus did not carry 2′OMe modifications than when it did so (Fig. 2e), an observation that deserves further investigation.

The improved tracrRNA design described here is likely to increase the editing efficiency of most targets and allow cleavage of otherwise non-editable loci. This will reduce the need to pre-screen several gRNAs to identify efficient ones for each target of interest. Stable hairpins can also be easily added to CRISPR sgRNA library designs and thus increase the efficiency of CRISPR screens. When the GOLD-tracrRNA will be used for editing of targets that have a strong tendency to generate off-targets, we suggest that a high fidelity Cas9 variant is used to prevent off-target editing (Supplementary Fig. 5). Because gRNA misfolding will not only prevent cleavage but also binding of Cas9 to the target (Supplementary Fig. 6), the GOLD-tracrRNA scaffold described may be beneficial also for applications in which Cas9 variants are linked to gene activators and repressors or base editors[26]. It could furthermore be applied to inherently self-complementary prime editing gRNAs[27] or to gRNAs of other CRISPR systems in which a hairpin is amenable to modification.

## Methods

**Cell culture**. We used a iCRISPR-Cas9 line derived from 409-B2 human induced pluripotent stem cells[13] (GMO permit AZ 54-8452/26), HEK293 cells (ECACC, 85120602) grown in DMEM/F-12 (Gibco, 31330-038) supplemented with 10% fetal bovine serum (FBS) (SIGMA, F2442) and 1x NEAA (SIGMA, M7145); K562 cells (ECACC, 89121407) grown in Iscove's modified Dulbecco's media (ThermoFisher, 12440053) supplemented with 10% FBS; Jurkat cells (ATCC, TIB-152) grown in RPMI 1640 medium (Gibco, 21875034) supplemented with 10% FBS; Caco-2 cells (DSMZ, ACC169) grown in MEM Eagle (SIGMA, M4655) supplemented with 20% FBS and 1× NEAA; SNL 76/7 (ECACC, 07032801) grown in KnockOut DMEM (Gibco, 10829018) supplemented with 2 mM GlutaMAX (Gibco, 35050061) and 10% FBS; as well as CHO-K1 cells (ATCC, CCL61) grown in Ham's F-12K medium (Gibco, 21127022) supplemented with 10% FBS. All cell lines were authenticated by the supplier via certificate of analysis and additionally in-house by checking morphology. All cell lines were tested negative for mycoplasma contamination before and after the experiments. 409-B2 hiPSCs were grown on Matrigel Matrix (Corning, 35248) in mTeSR1 medium (StemCell Technologies, 05851) with supplement (StemCell Technologies, 05852) and MycoZap Plus-CL (Lonza, VZA-2011) that was replaced daily. Medium for other cells was replaced every second day. At ~80% confluency, adherent cells were dissociated using EDTA (VWR, 437012 C) and split 1:6 to 1:10 in medium supplemented with 10 µM Rho-associated protein kinase (ROCK) inhibitor Y-27632 (Calbiochem, 688000) for one day after replating. K562 and Jurkat cells were split by 1:6 to 1:10 dilution after 1 week. Cells were grown at 37 °C in a humidified incubator with 5% CO$_2$. For experiments in which Cas9 should be produced by 409-B2 iCRISPR hiPSCs 2 µg/ml doxycycline (Clontech, 631311) was added to the media three days before transfection. For generation of single cell-derived cellular colonies (Fig. 3b), cells were dissociated and thoroughly separated using TrypLE Express (ThermoFisher, 12605036) and seeded in different dilutions.

**Production of sgRNA**. Secondary structure prediction of gRNAs was done using the RNAstructure web server[23]. Chemically synthesized crRNAs and tracrRNAs as

well as oligonucleotides for sgRNA production were from IDT (Coralville, USA) (Supplementary Data). For the production of sgRNAs by in vitro transcription (IVT) ssDNA templates were designed to contain the T7 promoter sequence in front of the sgRNA coding sequence. These ssDNA templates were hybridized with short complementary ssDNA for 2 min at 95 °C, cooled for 10 min at 20 °C to form a dsDNA T7 promoter and used for IVT according to the manufacturer's protocol (T7 High Yield RNA synthesis kit NEB, E2040S). The IVT reaction mix was incubated overnight at 37 °C before 2 μL of TURBO DNase (Invitrogen, AM2238) was added and incubation continued for 30 additional minutes. IVT products were purified with the MEGAclear Transcription clean-up kit (Invitrogen, AM1908). For experiments in which sgRNAs are expressed in the cell, dsDNA with a U6 promoter sequence in front of the sgRNA coding sequence were designed and amplified (98 °C 30 s; 35 × [98 °C 10 s, 61 °C 20 s, 72 °C 25 s]; 72 °C 5 min) using Phusion HF MasterMix (Thermo Scientific, F-531L). Amplifications were analyzed using EX agarose gels (Invitrogen, G4010-11) and PCR products were purified using solid phase reversible immobilization (SPRI) beads[28].

**Production of modified plasmids**. pSpCas9(BB)-2A-GFP (PX458) (Addgene #48138) and pSpCas9(BB)-2A-Puro (PX459) V2.0 (Addgene #62988) were used as vectors to integrate sequences coding for full-length gRNAs carrying either the normal gRNA backbone or the locked gRNA backbone (Supplementary Data) using BbsI (ThermoScientific, FD1014) restriction digest and cloning as described by the depositor. Cloned plasmids were used for heat-shock transformation of E.coli strain Stbl3 (Invitrogen, C737303). Bacteria were streaked on ampicillin (100 μg/ml) containing LB-agar (SIGMA, L2897) plates and incubated overnight at 37 °C, yielding single colony density. Single colonies were used to inoculate overnight cultures in LB-medium (Sigma, L3022) followed by plasmid DNA extraction with the Plasmid Plus Midi Sample Kit (Qiagen, 12941). To exclude unintended plasmid mutations that differ between plasmids modified to express the gRNAs with either the normal or locked backbone we used aliquots of purified plasmids for plasmid library preparation and sequencing. First, plasmids were sonicated three times with a Bioruptor (Diogenode), with the output selector switched to (H)igh to yield fragments of ~0.15 to 0.8 kb. The sheared fragments were blunted for 30 min at 20 °C with the Quick Blunting Kit (New Engand Biolabs, E1201L), purified with SPRI beads[28], adapter-ligated for 30 min at 20 °C with the Quick Ligation Kit (New England Biolabs, M2200L), again purified with SPRI beads, double-indexed by PCR-amplification and finally purified with SPRI beads. After Illumina sequencing, we compared the mapped sequences of the modified plasmids with the respective plasmid reference sequences using the integrative genomics viewer (IGV)[29].

**Lipofection of oligonucleotides**. Lipofection (reverse transfection) was done using the alt-CRISPR manufacturer's protocol (IDT) with a final concentration of 7.5 nM of each gRNA that had been previously formed by hybridization of crRNA and tracrRNA. In brief, 0.75 μl RNAiMAX (Invitrogen, 13778075) and the respective gRNAs were separately diluted in 25 μl OPTI-MEM (Gibco, 1985-062) each and incubated at 20 °C for 5 min. Both dilutions were mixed to yield 50 μl of OPTI-MEM including RNAiMAX and gRNAs. The lipofection mix was incubated for 20–30 min at 20 °C. 409-B2 iCRISPR hiPSCs were dissociated using EDTA for 5 min and counted using the Countess Automated Cell Counter (Invitrogen). The lipofection mix, 100 μl containing 25,000 dissociated cells in mTeSR1 supplemented with Y-27632 and 2 μg/ml doxycycline were thoroughly mixed and transferred to 1 well of a 96-well plate covered with Matrigel Matrix (Corning, 35248). Media was changed to regular mTeSR1 media after 24 h.

**Electroporation of oligonucleotides and RNPs**. Electroporation of oligonucleotides was done using the B-16 program of the Nucleofector 2b Device (Lonza) in cuvettes for 100 μl Human Stem Cell nucleofection buffer (Lonza, VVPH-5022), containing 1 million cells, 78 pmol electroporation enhancer, and 320 pmol of gRNA. For RNP based editing either 252 pmol S.p. HiFi Cas9 (IDT) or 252 pmol S.p. Cas9 (IDT) were used. For generation of 'worst-target' inserted cells 200 pmol of single-stranded DNA donor (Supplementary Data) was additionally added to the nucleofection buffer. When plasmids px458 or px459 were used for editing 5 μg of the respective plasmids was added to the nucleofection buffer. No doxycycline was added to the media of cells that were used for RNP or plasmid-based editing. For generation of HDR-edited cells, the small molecule M3814 (MedChemExpress, HY-101570) was added to the media for two days post-electroporation to increase HDR efficiency[30].

**Illumina library preparation and sequencing**. At least five days after transfection cells were detached using TrypLE (ThermoFisher, 12605036), pelleted, and resuspended in 15 μl QuickExtract (Lucigen, QE0905T). Incubation at 65 °C for 10 min, 68 °C for 5 min, and 98 °C for 5 min was performed to yield single stranded DNA. Primers containing adapter overhangs for Illumina sequencing (Supplementary Data) were used to amplify each locus in a T100 Thermal Cycler (Bio-Rad) using the KAPA2G Robust PCR Kit (SIGMA, KK5024) with buffer B and 3 μl of cell extract in a total volume of 25 μl. The thermal cycling profile of the PCR was: 95 °C 3 min; 34× (95 °C 15 s, 65 °C 15 s, 72 °C 15 s); 72 °C 60 s. P5 and P7 Illumina adapters with sample specific indices were added in a subsequent PCR reaction[31]

using Phusion HF MasterMix (Thermo Scientific, F-531L) and 0.3 μl of the first PCR product. The thermal cycling profile of the second PCR was: 98 °C 30 s; 25× (98 °C 10 s, 58 °C 10 s, 72 °C 20 s); 72 °C 5 min. Amplifications were verified by size separating agarose gel electrophoresis using 2% EX gels (Invitrogen, G4010-11). The indexed amplicons were purified using solid phase reversible immobilization (SPRI) beads[28]. Double-indexed libraries were sequenced on a MiSeq (Illumina) yielding 2 ×150 bp (+7 bp index). After base calling using Bustard (Illumina) adapters were trimmed using leeHom[32].

**Amplicon sequence analysis**. Bam-files were demultiplexed and converted into fastq files using SAMtools[33]. CRISPResso[34] was used to analyze fastq files for percentage of wild type and indel sequences. For analysis of precise insertion frequency of 'worst-case' targets in single cell-derived cellular clones, the expected amplicon was provided to CRISPResso to call the HDR event. Analysis was restricted to amplicons with a minimum of 70% similarity to the wild type sequence and to a window of 20 bp from each gRNA. Unexpected substitutions were ignored as putative sequencing errors.

**In vitro binding and cleavage assay**. The in vitro cleavage of target DNA was adapted from Zetsche et al.[35]. In brief, a ribonucleoprotein (RNP) mix consisting of 340 nM gRNA (hybridized crRNA/tracrRNA) and 165 nM Cas9 or dead (d)Cas9 (both IDT) was incubated for 20 min at 20 °C, and subsequently mixed with 75 nM of the corresponding dsDNA targets, that had been previously generated by hybridizing complementary ssDNA (Supplementary Data) for 2 min at 95 °C followed by 10 min at 20 °C. Samples were incubated at 37 °C for 16 h and then subjected to Proteinase K or RNaseA (in combination, alone or none of each). In case of both treatments, Proteinase K was done prior to RNase A digest. Proteinase K (NEB P8107S, 800 U/ml) digest was carried out at 50 °C for 30 min and RNaseA (Qiagen 1007885, 2 mg/ml) digest was performed at 37 °C for 30 min. Samples were subjected to size separating agarose gel electrophoresis using 4% EX gels (Invitrogen, G401004) for qualitative analysis as well as to capillary gel electrophoresis with a High Sensitivity DNA chip (Agilent, 5067-4626) for quantitative analysis of cleaved and uncleaved, as well as bound and unbound, target DNA.

**Statistics and reproducibility**. Bar graphs in figures were plotted and SEM error bars were calculated using GraphPad Prism 6 software. The number of replicates is stated in the respective figure legends. No statistical method was used to pre-determine sample size. The experiments were not randomized. Samples were prepared unblinded but in parallel. Analysis was performed based on numerical sample names, without the identity of the samples known during the analysis.

**Reporting summary**. Further information on research design is available in the Nature Research Reporting Summary linked to this article.

## Data availability

Source Data are provided with this paper. The NGS data generated in this study have been deposited in the Dryad database under accession code dryad.2rbnzs7mk. Data are also available on request from authors. Source data are provided with this paper.

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

## Acknowledgements

We thank Dominik Macak for help with cell culture and sequencing library preparation, Antje Weihmann and Barbara Schellbach for DNA sequencing, and András Tálas for providing experimentally tested refractory target sequences. pSpCas9(BB)-2A-GFP (PX458) and pSpCas9(BB)-2A-Puro (PX459) V2.0 were a gift from Feng Zhang (Addgene plasmids #48138 and #62988). Funding was provided by the Max Planck Society and the NOMIS foundation.

## Author contributions

S.R. and N.H. conceived the idea. S.R. and P.K. performed the experiments and analyzed the data. T.M. provided low performance ClinVar target sequences. S.R. wrote the paper with input from all authors.

## Funding

## Competing interests

A related patent application on improved gRNA design has been filed (patent applicant: Max Planck Society, inventors: S.R., N.H. and T.M., application number: EP21176366.9, status: pending). The remaining authors declare no competing interests.
