## [Peer Review File · Nature Communications]

Reviewers' Comments:

Reviewer #1:

Remarks to the Author:

In the present manuscript entitled "Improved gRNA secondary structures allow editing of target sites resistant to CRISPR-Cas9 cleavage", the authors aimed to solve the problem of inefficiency of targeting sites due to gRNA misfolding. The authors enhanced the stability of the gRNA secondary structure by adding additional sequences to extend a hairpin structure on the gRNA, which significantly enhanced the efficiency of some difficult-to-edit sites. Meanwhile, the editing efficiency was further improved by combining chemical modification of gRNA, and it was ordered as GOLD-tracrRNA. In general, this study is interesting and provide useful information in the research field of optimizing the CRISPR/Cas system. Nevertheless, several questions should be addressed before publication.

Comments/Suggestions,

1. The whole studies only used hiPSCs. I would like to see if the t-lock sgRNAs could increase efficiencies in HEK293T cells with px458 vector; if the t-lock sgRNAs and GOLD tracrRNAs could increase efficiencies in mouse embryos.
2. The locked tracrRNA did not increase editing efficiencies for two of ten tested sites. What is the reason? What kinds of targets that the locked tracrRNAs could not increase efficiencies?
3. The draft manuscript mainly selects some sites that may be inefficient due to gRNA misfolding, it is suggested to also randomly select at least 5 sites with high efficiency to study whether GOLD-tracrRNA can also maintain high efficiency.
4. Off-targeting is always a concern. When the structural stability of gRNA is improved, does off-targeting efficiency increase? I suggested to select 2 gRNAs to test.
5. In GOLD-tracrRNA, was the extended Lock RNA also chemically modified? I did not see the description in the text.
6. In Figure 1D, the predicted results of t0 structure folding are shown. The prediction of t-lock gRNA folding should be also shown for comparison.
7. In Figure 1E, "all results from (C)" should be "all results from (D)".

Reviewer #2:

Remarks to the Author:

Riesenberg et al present a considered and carefully executed study that aims to improve gene editing by stabilising CRISPR-Cas9 RNA structures to prevent unwanted structural rearrangements that can occur with certain spacer sequences. Their combination of stable hairpin structures combined with RNA modifications can make in some cases striking improvements in gene editing efficiency, but in other cases had more modest effects or little or no effect. The changes are not detrimental in the targets they explored, so one could argue that these changes are generally useful.

What is missing here is any analysis of the effects of their changes on the RNA structures themselves, using biophysical techniques to show whether they are more stable or are bound more stably by the Cas9. As it stands there are some interesting phenomenological observations but lacking either a wider genome analysis or a more structural proof of the role of the changes they employ. As it stands it is more suited to a specialised publication.

Point-by-point response:

Reviewer #1:

In the present manuscript entitled “Improved gRNA secondary structures allow editing of target sites resistant to CRISPR-Cas9 cleavage”, the authors aimed to solve the problem of inefficiency of targeting sites due to gRNA misfolding. The authors enhanced the stability of the gRNA secondary structure by adding additional sequences to extend a hairpin structure on the gRNA, which significantly enhanced the efficiency of some difficult-to-edit sites. Meanwhile, the editing efficiency was further improved by combining chemical modification of gRNA, and it was ordered as GOLD-tracrRNA. In general, this study is interesting and provide useful information in the research field of optimizing the CRISPR/Cas system. Nevertheless, several questions should be addressed before publication.

Comments/Suggestions,

1. The whole studies only used hiPSCs. I would like to see if the t-lock sgRNAs could increase efficiencies in HEK293T cells with px458 vector; if the t-lock sgRNAs and GOLD tracrRNAs could increase efficiencies in mouse embryos.

We have now added data of editing with the different gRNA backbones in four other human cell lines (HEK293, K562, Jurkat, and Caco-2), in SNL 76/7 which is derived from mouse embryonic fibroblasts, as well as in CHO-K1 derived from Chinese hamster ovary cells (end of 2nd paragraph on page 5 and Supplementary Figure 4).

In an extreme case in CHO-K1 cells the editing efficiency increased 1006-fold from 0.08% to 80.5% with the GOLD-tracrRNA. For the other targets in non-pluripotent cells, the GOLD-tracrRNA increased editing efficiencies 11.9-fold (range 3.5-26.8 -fold) relative to the tracrRNA without any improvements and 3.1-fold (range 1.4-7.4 -fold) relative to the locked tracrRNA without chemical modifications.

We also inserted gRNAs carrying the unmodified or locked backbone in the px458 (Cas9-2A-GFP) or px459 (Cas9-2A-*Puro*) plasmids and used them for editing of the SNL 76/7 mouse cell line. The locked backbone increased efficiency 1.4-2.6-fold compared to the unmodified backbone (Supplementary Fig. 1 updated panel B).

2. The locked tracrRNA did not increase editing efficiencies for two of ten tested sites. What is the reason? What kinds of targets that the locked tracrRNAs could not increase efficiencies?

We do not know why two sites (Fig. 1D, *PIGZ* and *SLITRK1*) did not result in increased editing efficiencies, but it is interesting that the commercial tracrRNA from IDT with proprietary chemical modifications was also unable to increase editing efficiencies for those sites. When considering the unmodified tracrRNA (t0), *PIGZ* and *SLITRK1* already had the highest editing efficiencies (>10%) in the screen of 10 sites using gRNA lipofection in 409-B2 hiPSCs.

3. The draft manuscript mainly selects some sites that may be inefficient due to gRNA misfolding, it is suggested to also randomly select at least 5 sites with high efficiency to study whether GOLD-tracrRNA can also maintain high efficiency. 4. Off-targeting is always a concern. When the structural stability of gRNA is improved, does off-targeting efficiency increase? I suggested to select 2 gRNAs to test.

It is important that efficient gRNAs maintains their efficiency and we agree that off-target editing could be a potential caveat when using very efficient gRNAs. We chose to address points 3) and 4) with a combined experimental setup. We selected 6 gRNAs based on their predicted tendency to result in off-target editing based on the CFD off-target score (Doench et al. 2016, Haeussler et al. 2016). As the gRNA spacers were selected based on off-target scores there should be no strong selection bias for low or high on-target efficiency. We then sequenced the on-target site, as well as the top two predicted off-target sites for each of the 6 gRNAs after editing using either the unmodified tracrRNA or the GOLD-tracrRNA. We write on page 5:

“Next, we investigated whether more efficient editing efficiency at the intended (on-) target by the GOLD-tracrRNA might result in increased editing of unintended targets that have sequence similarity to the on-target sit. We chose six gRNA spacers that differ in their predicted tendency to result in off-target editing based on the cutting frequency determination (CFD) score (Doench et al. 2016,

Haeussler et al. 2016) which ranges from 0 (unlikely off-target) to 1 (very likely off-target). The off-target scores of our gRNAs range from 0.2 to 1 (Supplementary Fig. 5A). After Cas9 RNP editing in 409-B2 hiPSCs with crRNA hybridized to either the unmodified tracrRNA or the GOLD-tracrRNA, we sequenced the on-target site as well as the top two predicted off-target sites for each of the six gRNAs. The GOLD-tracrRNA increased on-target editing efficiency for the six spacers relative to the unmodified tracrRNAs, but also increased the efficiency of editing of the two off-target sites with the highest CFD scores (0.91 and 1) from around 0.5% to 9% (Supplementary Fig. 5B). Another off-target with a high CFD score of 0.72 increased from 0.15% to 0.26%. The other nine off-target sites with CFD scores ranging from 0.2 to 0.78 were not edited more efficiently by the GOLD-tracrRNA. Importantly, when we edited the same targets using a high fidelity version of Cas9 with the R691A mutation (HiFi Cas9) (Vakulskas et al. 2018) the GOLD-tracrRNA increased on-target editing for all six spacers, while off-target editing at all 12 off-target sites was unchanged with respect to editing with the unmodified tracrRNA (Supplementary Fig. 5C).”

We also add to the discussion (page 8):

“When the GOLD-tracrRNA will be used for editing of targets that have a strong tendency to generate off-targets, we suggest that a high fidelity Cas9 variant is used to prevent off-target editing (Supplementary Fig. 5).”

5. In GOLD-tracrRNA, was the extended Lock RNA also chemically modified? I did not see the description in the text.

The locked hairpin is not chemically modified. This is now clarified in the text at the beginning of page 5.

6. In Figure 1D, the predicted results of t0 structure folding are shown. The prediction of t-lock gRNA folding should be also shown for comparison.

Folding predictions for the gRNAs from Fig.1 are identical except for an elongated 1st hairpin when the locked tracrRNA is chosen as input for the prediction. The only exception is the formation of a 1st hairpin in the case of *OSBP2* and *C3*, for which a hairpins are not predicted with the unmodified tracrRNA (see below). We have added a sentence describing this in the legend of Figure 1.

7. In Figure 1E, "all results from (C)" should be "all results from (D)".

Thank you for pointing this out. We have corrected the mistake.

Reviewer #2:

Riesenberg et al present a considered and carefully executed study that aims to improve gene editing by stabilising CRISPR-Cas9 RNA structures to prevent unwanted structural rearrangements that can occur with certain spacer sequences. Their combination of stable hairpin structures combined with RNA modifications can make in some cases striking improvements in gene editing efficiency, but in other cases had more modest effects or little or no effect. The changes are not detrimental in the targets they explored, so one could argue that these changes are generally useful.

What is missing here is any analysis of the effects of their changes on the RNA structures themselves, using biophysical techniques to show whether they are more stable or are bound more stably by the Cas9.

To assess whether changes in the gRNA structure change the amount of ribonucleoproteins competent to bind double-stranded target DNA, and/or increases cleavage efficiency, we have added two sets of experiments; (1) electrophoretic mobility shift assays using catalytically inactive ('dead') Cas9; (2) *in-vitro* cleavage assays. We have now added a separate subsection describing these experiments (page 6-7).

As it stands there are some interesting phenomenological observations but lacking either a wider genome analysis or a more structural proof of the role of the changes they employ. As it stands it is more suited to a specialised publication.

In addition to the *in vitro* binding and cleavage assay, we have added new data that provides a 'wider genome analysis' (see response to reviewer 1), specifically: (1) comparison of unmodified and improved gRNAs for nine new spacer sequences, included editing in four other human cell lines (HEK293, K562, Jurkat, and Caco-2), one mouse line (SNL 76/7) and one hamster line (CHO-K1) (end of 2nd paragraph on page 5 and Supplementary Figure 4). We also analyzed off-target efficiency at 12 sites (page 5 and Supplementary Fig. 5). In total, we now analyze the editing efficiency of 30 different gRNA spacer sequences using unmodified and improved gRNA designs.

Reviewers' Comments:

Reviewer #1:

Remarks to the Author:

All my concerns have been addressed, I have no more questions

Reviewer #2:

Remarks to the Author:

Riesenberg et al have provided an extensive response to my review and that of the other referee. They have covered my concerns by the additional data; in particular that off-target data will be of interest and is information of the mechanism via increased binding, which the bandshifts seem to confirm.

A small comment is that the typical abbreviation is tracrRNA. I didn't notice this before.

Point-by-point response:

Reviewer #1:

All my concerns have been addressed, I have no more questions

Thank you again for your valuable suggestions in the review process.

Reviewer #2

Riesenberg et al have provided an extensive response to my review and that of the other referee. They have covered my concerns by the additional data; in particular that off-target data will be of interest and is information of the mechanism via increased binding, which the bandshifts seem to confirm.

A small comment is that the typical abbreviation is tracrRNA. I didn't notice this before.

Thank you for pointing this out. We have corrected this mistake in the revised manuscript.